# Self-Supervised State-Control through Intrinsic Mutual Information Rewards

## Abstract

Learning useful skills without a manually-designed reward function would have many applications, yet is still a challenge for reinforcement learning. In this paper, we propose Mutual Information-based State-Control (MISC), a new self-supervised Reinforcement Learning approach for learning to control states of interest without any external reward function. We formulate the intrinsic objective as rewarding the skills that maximize the mutual information between the context states and the states of interest. For example, in robotic manipulation tasks, the context states are the robot states and the states of interest are the states of an object. We evaluate our approach for different simulated robotic manipulation tasks from OpenAI Gym and a navigation task in the Gazebo simulator. We show that our method is able to learn to manipulate the object, such as pushing and picking up, purely based on the intrinsic mutual information rewards. Furthermore, the pre-trained policy and mutual information discriminator can be used to accelerate learning to achieve high task rewards. Our results show that the mutual information between the context states and the states of interest can be an effective ingredient for overcoming challenges in robotic manipulation tasks with sparse rewards. A video showing experimental results is available at `https://youtu.be/l5KaYJWWu70`.

## 1 Introduction

Reinforcement Learning (RL) (Sutton & Barto, 1998) combined with Deep Learning (DL) (Goodfellow et al., 2016) has led to great successes in various reward-driven tasks, such as playing video games (Mnih et al., 2015), learning continuous control (Ng et al., 2006; Peters & Schaal, 2008; Levine et al., 2016; Chebotar et al., 2017), navigating in complex environments (Mirowski et al., 2017; Zhu et al., 2017), and manipulating objects (Andrychowicz et al., 2017; 2018).

Despite these successes, RL agents that learn only from reward signals differ in the manner that humans learn. In the case of learning to manipulate objects, a human agent not only attempts to accomplish the task but also learns to master the controllable aspects of the environment (Lake et al., 2017). Notably, a human agent can quickly learn the correlation between its own action and the state change of an object, even without supervision, to later use the acquired skill to manipulate the object into the desired state.

The ability to fully autonomously learn to control the states of interest has many benefits. First, it would make learning possible in the absence of hand-engineered reward functions or manually-specified agent goals. It is known that designing a reward function that ensures the agent to learn the desired behaviors is challenging (Hadfield-Menell et al., 2017). Secondly, to learn to "master" the environment potentially helps the agent to learn to achieve goals in sparse reward settings. Thirdly, the policy of controlling states of interest can be quickly adapted to unknown tasks. It is currently an open challenge to design RL agents that automatically learn useful skills to control the states of the environment, without rewards (Warde-Farley et al., 2019).

In this paper, a *skill* is a policy that changes the state of the environment in a consistent way. The policy can be a single unconditioned policy (Peters & Schaal, 2008) or a latent-condionted policy (Eysenbach et al., 2019). One way to learn skills is to simultaneously train a discriminator to discern the skill-options of the agent (Szepesvari et al., 2014) based on the states of the trajectory (Eysenbach et al., 2019). In this way, the agent should be able to learn a diverse set of skills.

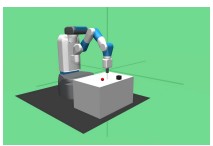 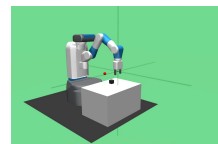 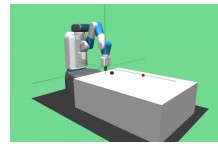 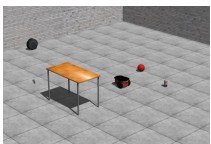

Figure 1: Fetch robot arm environments and a navigation task based on the Gazebo simulator: `FetchPush`, `FetchPickAndPlace`, `FetchSlide`, `SocialBot-PlayGround`.

Another way is to learn an environment model to encourage the agent to explore the states, which are relatively unpredictable (Houthooft et al., 2016; Pathak et al., 2017; Ha & Schmidhuber, 2018). These methods are more focused on efficiently exploring novel states instead of controlling states of interest.

We propose a new self-supervised reinforcement learning method, Mutual Information-based State-Control (MISC), which is an approach that learns skills to control states of interest without any reward signal. *States of interest* are the states of the environment that we are interested in. *Context states* are the states of the environment excluding the states of interest. The states of interest and the context states are given by the user before training the model. The idea of our method is to encourage the agent to find skills that maximize the mutual information between the context states and the states of interest. We first divide the observation states into two sets, the context states and the states of interest. In the case of robotic manipulation tasks, see Figure 1, the context states are the robot states; the states of interest are the states of an object. During the learning process of the agent, a discriminator learns to evaluate the mutual information between the context states and the states of interest. The agent receives high intrinsic rewards from the discriminator when there is high mutual information. Our hypothesis is that if the agent learns to control the object, then the mutual information of the agent states and the object states should be relatively high. Interestingly, this automated RL scheme bears similarities to the self-supervised learning of feature representations in computer vision tasks, where a neural network is trained to predict a part of the image given another part of the same image (Doersch et al., 2015). During the process, the neural network captures the mutual information among different parts of the image. Similarly, in our work, we want to capture the mutual information among different sets of states.

This paper contains the following five contributions. First, we introduce a new self-supervised RL method, Mutual Information-based State-Control, for learning to control states of interest. Secondly, we evaluate the developed framework in the robotic simulations of OpenAI Gym and a navigation task in Gazebo and demonstrate that MISC enables the agent to learn skills, such as reaching, pushing, picking up, and sliding the object without rewards. Thirdly, we show that the pre-trained policy can be quickly adapted to the specific tasks with external sparse reward signal. Fourthly, the pre-trained mutual information discriminator also improves the learning process of the agent, either as intrinsic rewards or as priorities for experience replay, in addition to the sparse rewards. Finally, we show that the learned mutual information discriminator can be transferred among different tasks and still improves the performance.

## 2 PROBLEM FORMULATION

In the RL setup, an agent interacts with an environment. The environment is fully observable and includes a set of states $S$, a set of actions $A$, a distribution of initial states $p(s_0)$, transition probabilities $p(s_{t+1} \mid s_t, a_t)$, a reward function $r \colon S \times A \to \mathbb{R}$, and a discount factor $\gamma \in [0, 1]$. These components formulate a Markov decision process represented as a tuple, $(S, A, p, r, \gamma)$. A policy $\pi$ maps a state to an action, $\pi \colon S \to A$. The goal of the agent is to maximize the accumulated reward, i.e., the return, $R_t = \sum_{i=t}^{\infty} \gamma^{i-t} r_i$, over all episodes, which is equivalent to maximizing the expected return, $\mathbb{E}_{s_0}[R_0 | s_0]$.

We consider robotic manipulation tasks, like the robotic simulations provided by OpenAI Gym (Todorov et al., 2012; Plappert et al., 2018), where a robot arm attempts to manipulate an object to a goal position via pushing, picking & placing, or sliding, as shown in Figure 1. We also consider a navigation task, where the robot should navigate to the target, which is a ball, as shown in Figure 1. The states of the environment $s$ consist of positions, orientations, linear and angular velocities of all

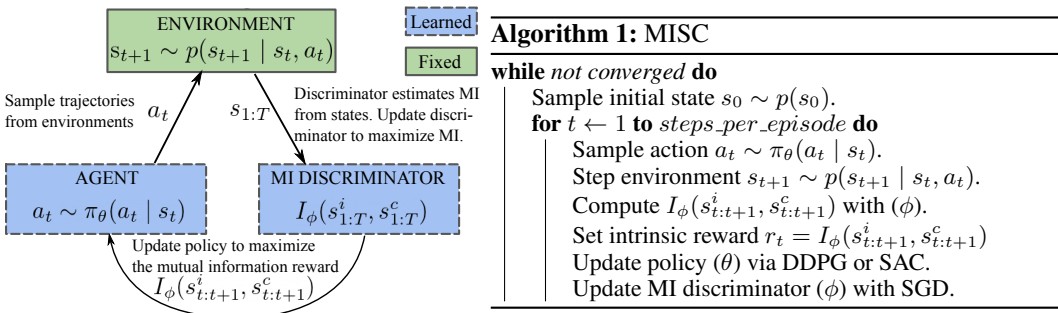

Figure 2: **MISC Algorithm**: We update the discriminator to better estimate the mutual information (MI), and update the agent to control states of interest to have higher MI with the context states.

robot joints and of an object. We divide the states $s$ into two sets, the robot states $s^r$ and the object states $s^o$. For manipulation tasks, we are interested in the states of the object. Therefore, we define the states of interest $s^i$ as the object states, $s^i = s^o$; the context states $s^c$ as the robot states, $s^c = s^r$. For simplicity, we represent the states of interest using only the object positions, $s^i = (x^o, y^o, z^o)$, and the context states using only the gripper positions, $s^c = (x^g, y^g, z^g)$.

In this paper, we focus on learning to control states of interest purely using the agent's observations and actions, without any reward function. Motivated by the idea that an agent capable of controlling states of interest $s^i$ to have high mutual information with its context state $s^c$ has "mastered" the environment, we formulate the problem of learning without external supervision as one of learning a policy $\pi_\theta(a_t \mid s_t)$ with parameters $\theta$ to maximize intrinsic mutual information rewards, $r = I(S^i; S^c)$. Here, we use upper letters, such as $S^i$, to denote random variables and the corresponding lower case letter, such as $s^i$, to represent the values of random variables.

## 3 METHOD

In this section, we formally describe our method, including the mutual information reward function and the self-supervised RL framework.

### 3.1 LEARNING A MUTUAL INFORMATION REWARD FUNCTION OF STATES

Our method simultaneously learns a policy and an intrinsic reward function by maximizing the mutual information between the states of interest and the context states. Mathematically, the mutual information between the state of interest random variable $S^i$ and the context state random variable $S^c$ is represented as:

$$\begin{aligned} I(S^i; S^c) &= H(S^i) - H(S^i \mid S^c) \\ &= D_{KL}(\mathbb{P}_{S^i S^c} \mid\mid \mathbb{P}_{S^i} \otimes \mathbb{P}_{S^c}) \end{aligned} \tag{1}$$

where $\mathbb{P}_{S^i S^c}$ is the joint probability distribution; $\mathbb{P}_{S^i} \otimes \mathbb{P}_{S^c}$ is the product of the marginal distributions $\mathbb{P}_{S^i}$ and $\mathbb{P}_{S^c}$; $D_{KL}$ denotes the Kullback-Leibler (KL) divergence. The first line in Equation (1) tells us that the agent should maximize the entropy of states of interest, $H(S^i)$, and concurrently, minimize the conditional entropy of states of interest given the context states, $H(S^i \mid S^c)$. When the conditional entropy, $H(S^i \mid S^c)$, is small, it becomes easy to predict the states of interest based on the context states. In this case, for example in robotic manipulation tasks, we can say that the robot can control the object because it is easy to predict the location of the object $S^i$, given the robot states $S^c$. The second line in Equation (1) gives us the mutual information in the KL divergence form.

Mutual information is notoriously difficult to compute in real-world settings (Hjelm et al., 2019). Motivated by Belghazi et al. (2018), we use a lower bound to approximate the mutual information quantity, $I(S^i; S^c)$. First, we rewrite the KL formulation of mutual information objective, Equa-

tion (1), using the Donsker-Varadhan representation (Donsker & Varadhan, 1975):

$$
\begin{aligned}
I(S^i; S^c) &= D_{KL}(\mathbb{P}_{S^i S^c} \| \mathbb{P}_{S^i} \otimes \mathbb{P}_{S^c}) \\
&= \sup_{\mathcal{T}:\Omega \to \mathbb{R}} \mathbb{E}_{\mathbb{P}_{S^i S^c}}[\mathcal{T}] - \log(\mathbb{E}_{\mathbb{P}_{S^i} \otimes \mathbb{P}_{S^c}}[e^{\mathcal{T}}])
\end{aligned}
\tag{2}
$$

where the input space $\Omega$ is a compact domain of $\mathbb{R}^d$, i.e., $\Omega \subset \mathbb{R}^d$; the supremum is taken over all functions $\mathcal{T}$ such that the two expectations are finite. Secondly, we lower bound the mutual information in Donsker-Varadhan representation with the compression lemma in the PAC-Bayes literature (Banerjee, 2006), mathematically,

$$
\begin{aligned}
I(S^i; S^c) &= \sup_{\mathcal{T}:\Omega \to \mathbb{R}} \mathbb{E}_{\mathbb{P}_{S^i S^c}}[\mathcal{T}] - \log(\mathbb{E}_{\mathbb{P}_{S^i} \otimes \mathbb{P}_{S^c}}[e^{\mathcal{T}}]) \\
&\geq \sup_{\phi \in \Phi} \mathbb{E}_{\mathbb{P}_{S^i S^c}}[\mathcal{T}_\phi] - \log(\mathbb{E}_{\mathbb{P}_{S^i} \otimes \mathbb{P}_{S^c}}[e^{\mathcal{T}_\phi}]) \\
&= I_\Phi(S^i, S^c).
\end{aligned}
\tag{3}
$$

The expectations in Equation (3) are estimated using empirical samples from $\mathbb{P}_{S^i S^c}$ and $\mathbb{P}_{S^i} \otimes \mathbb{P}_{S^c}$ or by shuffling the samples from the joint distribution along the batch axis (Belghazi et al., 2018). The mutual information reward function, $r = I_\Phi(S^i, S^c)$, can be trained by gradient ascent. The statistics model $\mathcal{T}_\phi$ is parameterized by a deep neural network with parameters $\phi \in \Phi$, which aims to estimate the mutual information with arbitrary accuracy.

## 3.2 MUTUAL INFORMATION-BASED STATE-CONTROL

We want to train an agent to "master" the states of interest in a self-supervised reinforcement learning fashion. The agent aims to maximize the mutual information between the states of interest and the context states. At the beginning of each episode, the agent takes actions $a_t$ following a partially random policy, such as $\epsilon$-greedy, to explore the environment and collects trajectories into a buffer for later replay. The trajectory contains a series of states, $\{s_1, s_2, \ldots, s_T\}$, where $T$ is the time horizon of the trajectory. Each state $s_t$ consists of the states of interest $s_t^i$ and the context states $s_t^c$, where in robot manipulation tasks, we use the object position as $s_t^i$, and the gripper position as $s_t^c$.

For training the mutual information discriminator network, we first randomly sample trajectories of states $s_{1:T} = \{s_1, s_2, \ldots, s_T\}$ from the replay buffer. Here, we use $s_{t:t'}$ to indicate a subsequence of a trajectory, for instance, $s_{1:t}$ refers to $\{s_1, s_2, \ldots, s_t\}$. Then, we evaluate the mutual information lower bound, $I_\phi(s_{1:T}^i, s_{1:T}^c)$, mathematically,

$$
I_\phi(s_{1:T}^i, s_{1:T}^c) = \frac{1}{T} \sum_{t=1}^{T} \mathcal{T}_\phi(s_t^i, s_t^c) - \log\left(\frac{1}{T} \sum_{t=1}^{T} e^{\mathcal{T}_\phi(s_t^i, \bar{s}_t^c)}\right),
\tag{4}
$$

where the states $\bar{s}_t^c$ are sampled by shuffling the states $s_t^c$ along the temporal axis $t$ within each trajectory. After evaluating the lower bound, $I_\phi(s_{1:T}^i, s_{1:T}^c)$, we use back-propagation to optimize the parameter ($\phi$) to maximize the mutual information lower bound. Here, we calculate the mutual information using the samples from the same trajectory. If the context states or the states of interest do not change within the trajectory, then the mutual information of these states is zero. Therefore, if the agent does not alter the object states during the episode, then the mutual information between the agent states and the object states remains zero.

We define the transition mutual information reward as the mutual information estimation of a trajectory fraction, $s_{t:t+1}$, mathematically,

$$
r_t = I_\phi(s_{t:t+1}^i, s_{t:t+1}^c) = \text{clip}\left(\frac{1}{2} \sum_{t=t}^{t+1} \mathcal{T}_\phi(s_t^i, s_t^c) - \log\left(\frac{1}{2} \sum_{t=t}^{t+1} e^{\mathcal{T}_\phi(s_t^i, \bar{s}_t^c)}\right), 0, I_{tran}^{max}\right),
\tag{5}
$$

where $I_{tran}^{max}$ is the predefined maximal transition mutual information value. The clip function limits the transition mutual information value in an interval of $[0, I_{tran}^{max}]$. The lower threshold 0 forces the mutual information estimation to be non-negative. In practice, to mitigate the influence of some particular large transition mutual information, we find it useful to clip the intrinsic reward with the threshold value $I_{tran}^{max}$. This clipping trick makes the training stable. The threshold value can be considered as a hyper-parameter.

We implement MISC with both deep deterministic policy gradient (DDPG) (Lillicrap et al., 2016) and soft actor-critic (SAC) (Haarnoja et al., 2018) to learn a policy $\pi_\theta(a \mid s)$ that aims to control the states of interest. In comparison to DDPG and SAC, the DDPG method improves the policy in a more "greedy" fashion, while the SAC approach is more conservative, in the sense that SAC incorporates an entropy regularizer $\mathcal{H}(A \mid S)$ that maximizing the policy's entropy over actions. We found empirically that DDPG works better when the agent starts learning quickly, while SAC explores the environment more thoroughly in general.

Overall, the agent is rewarded for controlling states of interest to have higher mutual information with its context states, which is considered to gain a degree of mastery of the environment. We summarize the complete training algorithm in Algorithm 1 and in Figure 2.

## 4 EXPERIMENTS

To evaluate MISC, we use the robotic manipulation environments provided by OpenAI Gym and also a navigation task, see Figure 1 (Brockman et al., 2016; Plappert et al., 2018). First, we analyze the skills learned purely with the intrinsic reward. We show that the agent is able to learn skills, such as reaching, pushing, picking up, and sliding an object, see Figure 3. We also compare our method with "Diversity is All You Need" (DIAYN) method (Eysenbach et al., 2019). Secondly, we show that the pre-trained models, including the policy and the MI discriminator, can be used for improving performance in conjunction with the rewards. There are several ways to improve performance, such as via policy initialization, with the MI intrinsic rewards, and using the MI value for prioritized experience replay. Interestingly, we also show that the pre-trained MI discriminator can be transferred among different tasks and still improves performance. We combine or compare our method with other methods, including DDPG (Lillicrap et al., 2016), SAC (Haarnoja et al., 2018), PPO (Schulman et al., 2017), DIAYN (Eysenbach et al., 2019), PER (Schaul et al., 2016), VIME (Houthooft et al., 2016), ICM (Pathak et al., 2017), and Empowerment (Mohamed & Rezende, 2015). Thirdly, we show some insights about how MISC rewards are distributed across a trajectory. The detailed experimental details are shown in the Appendix. Our code is available at https://github.com/misc-project/misc.

### 4.1 ANALYSIS OF LEARNED SKILLS

**Question 1.** *What skill does MISC learn?*

We tested MISC in multiple robotic manipulation tasks, including push, pick & place, and slide, see Figure 1. In the push and slide tasks, the gripper is fixed to be closed. In the pick & place task, the agent has the control of the gripper to be open or closed. Unlike in push and pick & place environments, the surface of the table in the slide task has very low friction so that the object can slide on these surface. In all the environments, the object is randomly placed on the table at the beginning of each episode. During training, the agent only receives the intrinsic MISC reward, see Section 3.2. In all three environments, the behavior of reaching objects emerges from the self-supervised training, for example, as shown in Figure 3 (1st row). Furthermore, in the push and pick & place environments, the agent learns to push the object around on the table, see Figure 3 (2nd row). In the slide environment, the agent learns to slide the object into different directions, see Figure 3 (3rd row). Perhaps surprisingly, in the pick & place environment, the agent learns to pick up the object from the table without any reward, which means taking full control of the object states, see Figure 3 (4th row).

We implemented the MISC with both DDPG and SAC and ran the experiments with 5 different random seeds. To compare DDPG+MISC and SAC+MISC, we ran 20 trials using the learned policy in the pick & place environment with each seed. We observed that, in all 5 random seed settings, SAC+MISC learns the picking-up skill, while DDPG+MISC learns to pick up an object in only 1 out of 5 random seed settings. Mostly, the agent learns to push, flip, or grip the object. These observations show that the entropy bonus, $\mathcal{H}(A \mid S)$, of SAC can incorporate with MISC and helps the agent to better explore the skill space.

**Question 2.** *Can we use learned skills to directly maximize the task reward?*

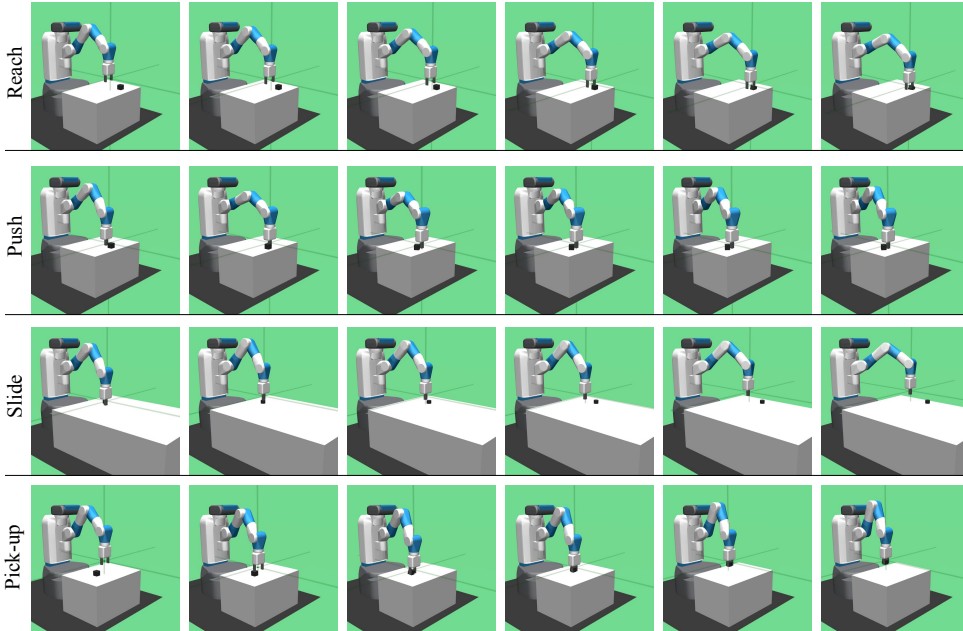

Figure 3: **Manipulation skills**: Without any reward, MISC learns skills for reaching, pushing, sliding, and picking up an object. The video of the learned skills is shown at `https://youtu.be/l5KaYJWWu70`. It is time-consuming to manually design reward functions that enforce these behaviors.

We tested our method, MISC, in the navigation task, `SocialBot-PlayGround`, which is based on the Gazebo simulator, as shown in Figure 1. In this navigation task, the external task reward is 1 if the agent reaches the ball, otherwise, the task reward is 0. We combine our method with PPO (Schulman et al., 2017)

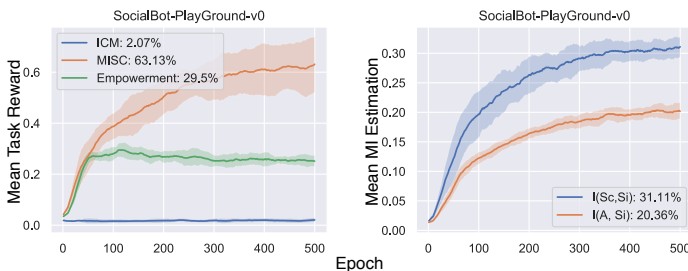

Figure 4: **Experimental results in the navigation task**

and compare the performance with ICM (Pathak et al., 2017) and Empowerment (Mohamed & Rezende, 2015). During training, we only use one of the intrinsic rewards such as MISC, ICM, or Empowerment to train the agent. Then, we use the averaged task reward as the evaluation metric. The experimental results are shown in FIgure 4 (left). The y-axis represents the mean task reward and the x-axis denotes the training epochs. From Figure 4 (left), we can see that the proposed method, MISC, has the best performance. The Empowerment has the second-best performance. The intuition why MISC, $I(S^c, S^i)$, is superior to Empowerment, $I(A, S^i)$, is that, the mutual information between the object states, $S^i$, and the robot states, $S^c$, is relatively high compared to the mutual information between agent's action $A$ and the state of interests $S^i$, as shown in Figure 4 (right). Subsequently, higher mutual information reward encourages the agent to explore more states, which have high mutual information. A theoretical connection between Empowerment and MISC is shown in the Appendix. Furthermore, the ICM method does not enable the agent to navigate to the ball because it only seeks novel states instead of controlling these states. A video showing the MISC-trained agent is available at `https://youtu.be/l5KaYJWWu70?t=104`.

#### Question 3. *How does MISC compare to DIAYN (Eysenbach et al., 2019)?*

The most related prior work on unsupervised skill learning, DIAYN, introduces an information-theoretic objective $\mathcal{F}_{\text{DIAYN}}$, which learns diverse discriminable skills indexed by the latent variable

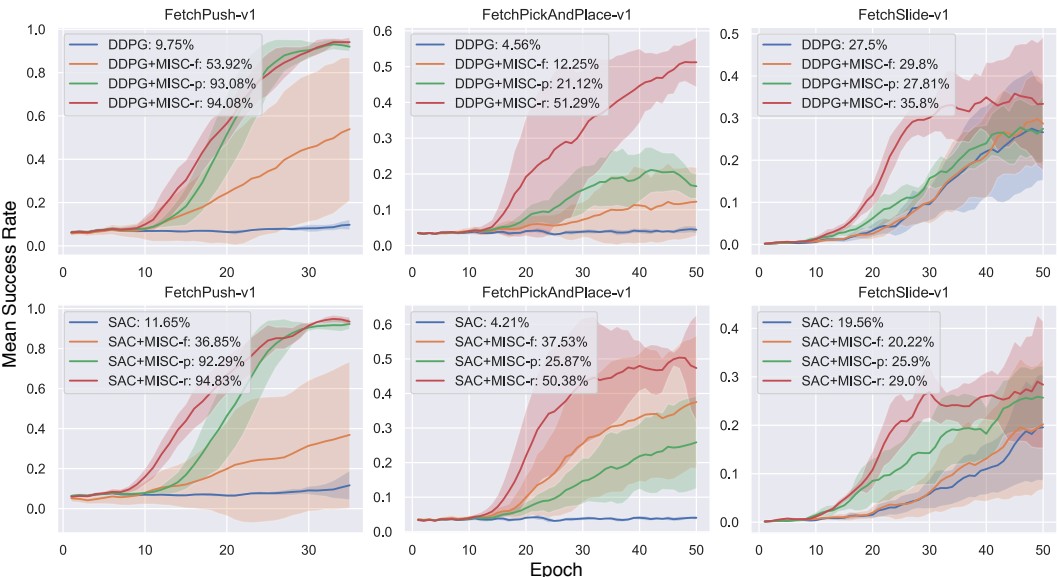

Figure 5: **Mean success rate with standard deviation:** The percentage values after colon (:) represent the best mean success rate during training. The shaded area describes the standard deviation.

$Z$, mathematically,

$$
\begin{aligned}
\mathcal{F}_{\text{DIAYN}} &= I(S; Z) + \mathcal{H}(A \mid S) - I(A; Z \mid S) \\
&= I(S; Z) + \mathcal{H}(A \mid S, Z) \\
&\geq \mathbb{E}_{z \sim p(z), s \sim \pi(z)}[\log q_\phi(z \mid s) - \log p(z)] + \mathcal{H}(A \mid S, Z).
\end{aligned}
\tag{6}
$$

The first term, $I(S; Z)$, in the objective, $\mathcal{F}_{\text{DIAYN}}$, is implemented via a skill discriminator, which is proven to be a variational lower bound of the original objective (Barber & Agakov, 2003; Eysenbach et al., 2019). The skill discriminator assigns high rewards to the agent, if it can predict the skill-options, $Z$, given the states, $S$. The second term, $\mathcal{H}(A \mid S, Z)$, is implemented through SAC (Haarnoja et al., 2018) conditioned on skill-options (Szepesvari et al., 2014). In the DIAYN paper, the authors mainly evaluated the algorithm on navigation and locomotion environments (Todorov et al., 2012; Brockman et al., 2016). We can adapt DIAYN to manipulation tasks by replacing the full states, $S$, with states of interest, $S^i$, i.e., the object states, as $I(S^i; Z)$. In comparison, our method MISC proposes to maximize the mutual information between the context states and the states of interest, i.e., $I(S^c; S^i)$, so that the agent learns to control the states of interest in a self-supervised fashion. These two methods can be combined as follows:

$$
\mathcal{F}_{\text{MISC+DIAYN}} = I(S^c; S^i) + I(S^i; Z) + \mathcal{H}(A \mid S, Z).
\tag{7}
$$

We compare MISC, DIAYN and MISC+DIAYN in the pick & place environment. For implementing MISC+DIAYN, we first pre-train the agent with only MISC, and then fine-tune the policy with DIAYN. After pre-training, MISC learns manipulation behaviors such as, reaching, pushing, sliding, and picking up, as shown in Figure 3. Compared to MISC, DIAYN rarely learns to pick up the object. We found that DIAYN does not fully control the object. The DIAYN-trained agent mostly pushes or flicks the object with the gripper. However, the combined model, MISC+DIAYN, learns to pick up the object and moves it to different locations, which depends on the conditioned skill-option. These observations are shown in the video at `https://youtu.be/l5KaYJWWu70?t=47`. MISC helps the agent to learn the DIAYN objective. The agent first learns to control the object with MISC, and then discovers diverse manipulation skills with DIAYN. The combination of MISC and DIAYN can be used for learning motion primitives via skill-conditioned policy for hierarchical reinforcement learning (Eysenbach et al., 2019).

## 4.2 ACCELERATING LEARNING WITH MISC

**Question 4.** *How can we use the learned skills or the trained MI discriminator to accelerate learning new tasks?*

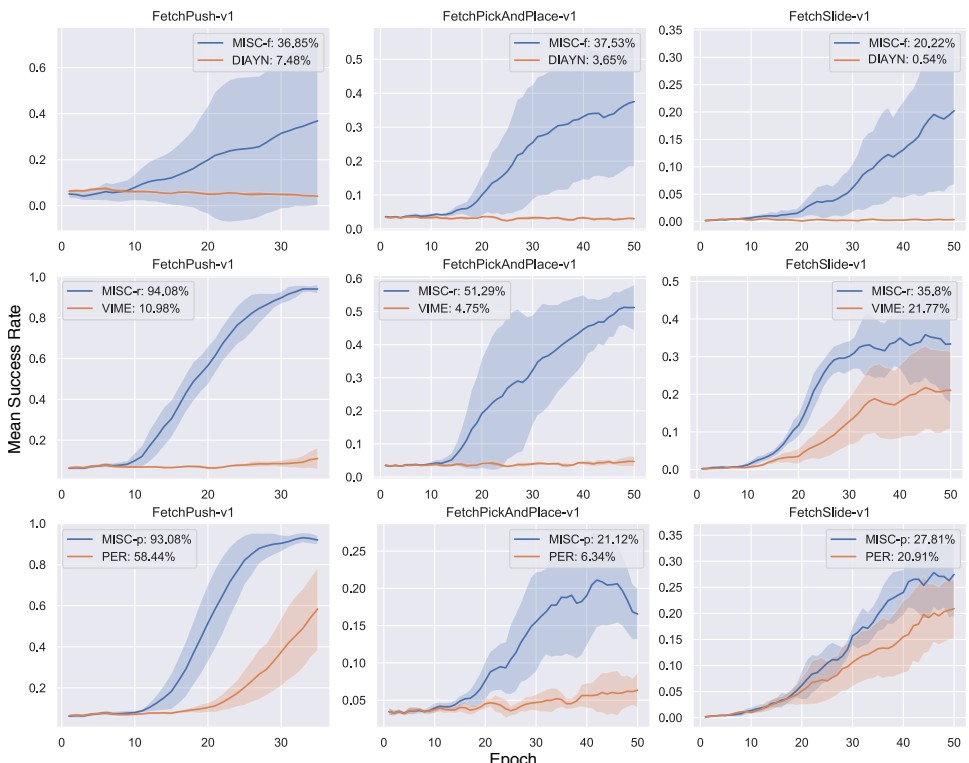

Figure 6: **Performance comparison:** We compare the MISC variants, including MISC-f, MISC-r, and MISC-p, with DIAYN, VIME, and PER, respectively.

We investigate three ways of using the pre-trained policy and the MI discriminator to accelerate learning in addition to the task reward.

The first method is using the MISC pre-trained policy as parameter initialization and fine-tune the agent with rewards. We denote this variant as "MISC-f", where "-f" stands for fine-tuning. The second variant is to use the MI intrinsic reward to help the agent explore high mutual information states. We name this method as "MISC-r", where "-r" stands for rewarding. The third approach is to use the mutual information quantity from MISC to prioritize trajectories for replay. We name this method as "MISC-p", where "-p" stands for prioritization.

We combined these three variants with DDPG and SAC and tested them in the robotic simulations provided in OpenAI Gym. The environments, including push, pick & place, and slide, have a set of predefined goals, which are represented as the red dots, as shown in Figure 1. The task for the RL agent is to manipulate the object to the goal positions. We ran all the methods in each environment with 5 different random seeds and report the mean success rate and the standard deviation, as shown in Figure 5. The percentage values alongside the plots are the best mean success rates during training. Each experiment is carried out with 16 CPU-cores.

From Figure 5, we can see that all these three methods, including MISC-f, MISC-p, and MISC-r, accelerate learning in the presence of task rewards. Among these variants, the MISC-r has the best overall improvements. In the push and pick & place tasks, MISC enables the agent to learn in a short period of training time, as shown in Figure 5. In the slide tasks, MISC-r also improves the performances by a decent margin.

We also compare our methods with more advanced RL methods. To be more specific, we compare MISC-f against parameter initialization using DIAYN (Eysenbach et al., 2019); MISC-p against Prioritized Experience Replay (PER), which uses TD-errors for prioritization (Schaul et al., 2016); MISC-r versus Variational Information Maximizing Exploration (VIME) (Houthooft et al., 2016). The experimental results are shown in Figure 6. From Figure 6 (1$^{st}$ row), we can see that MISC-f enables the agent to learn, while DIAYN does not. The reason is that DIAYN is more focused on

learning a diverse set of skills instead of learning to control the states. In the manipulation tasks, controlling states is more important, therefore MISC outperforms DIAYN. In the 2nd row of Figure 6, MISC-r performs better than VIME. This result indicates that the mutual information between states is a crucial quantity for accelerating learning. The mutual information intrinsic rewards boost performance significantly compared to VIME. This observation is consistent with the experimental results of MISC-p and PER, as shown in Figure 6 (3rd row), where the mutual information-based prioritization framework performs better than the TD-error based approach, PER. On all tasks, MISC enables the agent to learn the benchmark task more quickly.

## 4.3 TRANSFER LEARNING WITH MISC

**Question 5.** *Can the learned MI discriminator be transferred to different tasks?*

It would be beneficial if the pre-trained MI discriminator can be transferred to new tasks and still improves the performance (Pan et al., 2010; Bengio, 2012). To verify this idea, we directly applied the pretrained MI discriminator from the pick & place environment to the push and slide environments, respectively. We denote this transferred method as "MISC-t", where "-t" stands for

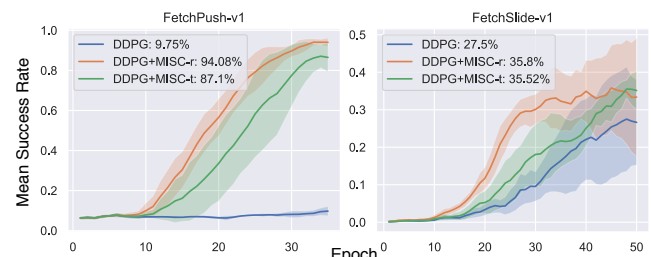

Figure 7: **Transferred MISC**

transfer. The MISC reward function trained in its corresponding environments is denoted as "MISC-r". We compare the performance of DDPG baseline, MISC-r, and MISC-t. The results are shown in Figure 7. Perhaps surprisingly, the transferred MISC still improves the performance significantly. Furthermore, as expected, MISC-r performs better than MISC-t in both tasks. We can see that the MI discriminator can be trained in a task-agnostic fashion and later utilized in unseen tasks.

## 4.4 INSIGHTS AND MORE

**Question 6.** *How does MISC distribute rewards across a trajectory?*

To understand why MISC works and how MISC distributes rewards, we visualize the learned MISC rewards, defined in Section 3.2, in Figure 8 and in a video at `https://youtu.be/l5KaYJWWu70?t=92`. From Figure 8, we can observe that the mutual information reward peaks between the fourth and fifth frame, where the robot quickly picks up the object from the table.

Around the peak reward value, the middle range reward values are corresponding to the relatively slow movement of the object and the gripper, see the third, ninth, and tenth frame. When there is no contact between the gripper and the object, see the first two frames in Figure 8, or the gripper holds the object still, see the sixth to eighth frames, the

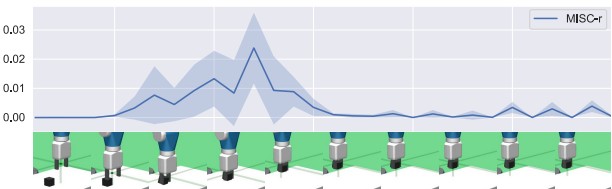

Figure 8: **MISC rewards over a trajectory**

intrinsic reward remains nearly zero. From this example, we see that MISC distributes positive intrinsic rewards, when the object states, $s^i$, has correlated changes with the robot state, $s^c$.

**Question 7.** *Can MISC help the agent to learn skills when there are no object states? And what happens if there are multiple objects?*

In the navigation task, we demonstrate that the agent learns to balance itself and run in a straight line, see video at `https://youtu.be/l5KaYJWWu70?t=134`, when the MISC objective is defined to maximize the mutual information between its left wheel and its right wheel.

When there are multiple objects of interest, for example a red and a blue ball on the ground. We can define the MISC objective as follows:

$$\mathcal{F}_{\text{MISC}} = I(S^c; S_1^i) + I(S^c; S_2^i). \tag{8}$$

With this objective, Equation (8), the agent learns to reach both balls and sometimes also learns to use one ball to hit the other ball. The results are shown in the video at `https://youtu.be/l5KaYJWWu70?t=148`.

From these examples, we can see that, with different pairs of state of interests and the context state, the agent is able to learn different kinds of skills. When there is no specific state of interests, we can train a skill-conditioned policy corresponding to different combinations of the two sets of states and later use the pre-trained policy for the tasks at hand.

## 5 RELATED WORK

Unsupervised skill learning is a challenging topic in RL. Variational Intrinsic Control (VIC) (Gregor et al., 2016) proposes an information-theoretic objective (Barber & Agakov, 2003) to jointly maximize the entropy of a set of options while keeping the options distinguishable based on the final states of the trajectory. Recently, Eysenbach et al. (2019) introduce DIAYN, which maximizes the mutual information between a fixed number of skill-options and the entire states of the trajectory. Eysenbach et al. (2019) show that DIAYN can scale to more complex tasks compared to VIC and provides a handful of low-level primitive skills as the basis for hierarchical reinforcement learning.

Another trunk of work on skill learning is based on goal-conditioned policies. Warde-Farley et al. (2019) propose, DISCERN, a method to learn a mutual information objective between the states and goals, which enables the agent to learn to achieve goals in environments with continuous high-dimensional observation spaces. Based on DISCERN, Pong et al. (2019) introduce Skew-fit, which adapts a maximum entropy strategy to sample goals from the replay buffer (Zhao & Tresp, 2019; Zhao et al., 2019) in order to make the agent learn more efficiently in the absence of rewards. More recently, Hartikainen et al. (2019) propose to automatically learn dynamical distances: a measure of the expected number of time steps to reach a given goal state, which can be used as rewards for learning to achieve new goals.

Similar to the skill learning methods, intrinsic rewards are also often used to help the agent learn more efficiently to solve tasks. For example, Jung et al. (2011) and Mohamed & Rezende (2015) use empowerment, which is the mutual information between states and actions, for intrinsically motivated agents. A theoretical connection between MISC and empowerment is shown in the Appendix. VIME (Houthooft et al., 2016) and ICM (Pathak et al., 2017) use curiosity as intrinsic rewards to encourage the agents to explore the environment more thoroughly.

Based on a similar motivation as previous works, we introduce MISC, a method that uses the mutual information between the states of interest and the context states (Doersch et al., 2015) as an intrinsic reward function. MISC enables the agent to learn manipulation skills without supervision. Our method is complementary to the previous skill learning works, such as DIAYN, and can be combined with them. The idea of MISC is to encourage the agent to learn to control states of interest, as one step forward towards mastery of the environment. Inspired by previous works (Schaul et al., 2016; Houthooft et al., 2016; Zhao & Tresp, 2018; Eysenbach et al., 2019), we additionally demonstrate three variants, including MISC-based fine-tuning, rewarding, and prioritizing mechanisms, to accelerate learning in the case when the task rewards are available.

## 6 CONCLUSION

This paper introduces Mutual Information-based State-Control (MISC), a self-supervised reinforcement learning framework for attaining useful skills. MISC utilizes a mutual information-based theoretic objective to encourage the agent to control states of interest without any reward function. In robotic manipulation tasks, MISC enables the agent to self-learn manipulation behaviors, like reaching, pushing, picking up, and sliding an object. In the navigation task, the MISC-trained agent learns to navigate to the target. Additionally, the unsupervised pre-trained policy and the mutual information discriminator accelerate learning in the presence of task rewards. We evaluate three MISC-based variants in different tasks and demonstrate a substantial improvement in learning efficiency compared to state-of-the-art methods.

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

# A    CONNECTION TO EMPOWERMENT

We divide the states $S$ into the states of interest $S^i$ and the context states $S^c$. In robot manipulation tasks, we define the states of interest as the object states, and the context states as the gripper states of the robot. The action space is the change of the gripper position and the status of the gripper, such as open or closed. Note that, the agent's actions directly control the context states.

Here, given the assumption that the transform, $S^c = F(A)$, from the actions, $A$, to the context states, $S^c$, is a smooth and uniquely invertible mapping (Kraskov et al., 2004), then we can prove that the MISC objective, $I(S^c, S^i)$, is equivalent to the empowerment objective, $I(A, S^i)$.

The empowerment objective (Klyubin et al., 2005; Salge et al., 2014; Mohamed & Rezende, 2015) is defined as the channel capacity in information theory, which means the amount of information contained in the actions $A$ about the states $S$, mathematically:

$$\mathcal{E} = I(S, A). \tag{9}$$

Here, we replace the states variable $S$ with sates of interest $S^i$, we have the empowerment objective as follows,

$$\mathcal{E} = I(S^i, A). \tag{10}$$

**Theorem 1.** *The MISC objective, $I(S^c, S^i)$, is equivalent to the empowerment objective, $I(A, S^i)$, given the assumption that the transform, $S^c = F(A)$, is a smooth and uniquely invertible mapping:*

$$I(S^c, S^i) = I(A, S^i) \tag{11}$$

where $S^i$, $S^c$, and $A$ denotes the states of interest, the context states, and the actions, respectively.

*Proof.*

$$
\begin{aligned}
I(S^c, S^i) &= \int \int ds^c ds^i p(s^c, s^i) \log \frac{p(s^c, s^i)}{p(s^c)p(s^i)} \\
&= \int \int ds^c ds^i \left\| \frac{\partial A}{\partial S^c} \right\| p(a, s^i) \log \frac{\left\| \frac{\partial A}{\partial S^c} \right\| p(a, s^i)}{\left\| \frac{\partial A}{\partial S^c} \right\| p(a)p(s^i)} \\
&= \int \int ds^c ds^i J_A(s^c) p(a, s^i) \log \frac{J_A(s^c)p(a, s^i)}{J_A(s^c)p(a)p(s^i)} \\
&= \int \int da ds^i p(a, s^i) \log \frac{p(a, s^i)}{p(a)p(s^i)} \\
&= I(A, S^i)
\end{aligned}
\tag{12}
$$

$\square$

## B  EXPERIMENTAL DETAILS

The experiments of the robotic manipulation tasks in this paper use the following hyper-parameters:

- Actor and critic networks: 3 layers with 256 units each and ReLU non-linearities
- Adam optimizer (Kingma & Ba, 2014) with $1 \cdot 10^{-3}$ for training both actor and critic
- Buffer size: $10^6$ transitions
- Polyak-averaging coefficient: 0.95
- Action L2 norm coefficient: 1.0
- Observation clipping: $[-200, 200]$
- Batch size: 256
- Rollouts per MPI worker: 2
- Number of MPI workers: 16
- Cycles per epoch: 50
- Batches per cycle: 40
- Test rollouts per epoch: 10
- Probability of random actions: 0.3
- Scale of additive Gaussian noise: 0.2

All hyper-parameters are described in greater detail at this link `https://github.com/misc-project/misc/tree/master/params`.

