# OpenReview forum: "Self-Supervised State-Control through Intrinsic Mutual Information Rewards"
_ICLR.cc/2020/Conference — Reject_

### Official Review · AnonReviewer3 · 2019-10-22
**Official Blind Review #3**

**Rating:** 6

**Review:**

I take issue with the usage of the phrase "skill discovery". In prior work (e.g. VIC, DIAYN), this meant learning a skill-conditional policy. Here, there is only a single (unconditioned) policy, and the different "skills" come from modifications of the environment -- the number of skills is tied to the number of environments. This is not to say that this way of doing things is wrong, but rather that it is misleading in the context of prior work. Skill discovery in this context implies being able to have a single agent execute a variety of learned skills, rather than having one agent per environment with each environment designed to elicit a specific skill.

Rather than "skill discovery", I suggest the authors position MISC relative to earlier work on empowerment, wherein a single policy was used to maximize mutual information of the form I(a; s_t | s_{t-1}). Modifying the objective to incorporate domain knowledge (as done in your DIAYN baseline) yields I(a; s_i | s_{t-1}) and is amenable to maximization by either of the lower bounds considered here. Indeed, your DIAYN baseline with skill length set to 1 and the number of skills equal to the number of actions (or same parameterization in the case of continuous actions) should recover this approach. I believe this would be a much more appropriate baseline, and I'd be curious to hear the intuition for why I(s_c ; s_i) should be superior.

Apart from this missing baseline, the experimental results seem convincing. However, it is unclear whether or not VIME and PER were modified to incorporate domain knowledge (i.e. s_i/s_c distinction). Indeed, an appendix would be greatly appreciated, as many experimental details were omitted. Ideally, an experimental setup with previously published results (e.g. control suite for DIAYN, Seaquest for DISCERN) would be considered, but I can understand why this wasn't done as incorporating domain knowledge is the main contribution of the paper. That said, the claims should be weakened to reflect this gap, and domain knowledge should be mentioned more prominently (e.g. states of interest vs context are given, not learned).

Rebuttal EDIT:

The language around skills and the extent of prior knowledge still downplays things a bit too much for my liking. Needing new environment variations to obtain new skills is a large step backwards from things like DIAYN (the MISC/DIAYN combination needs more evidence to be considered a possible solution), and the s_i/s_c distinction is non-trivial to specify or learn for harder problems (e.g. pixel observations).

That said, in the sort of settings under consideration (low dimensional state variables and environmental variations are simple to create) MISC does appear to be superior to prior work. The empowerment baseline is much appreciated, and while modifications of PER and VIME that incorporate prior knowledge would've also been nice, the experimental results pass the bar for acceptance in my view.



**Experience Assessment:**

I have published in this field for several years.

**Review Assessment: Checking Correctness Of Derivations And Theory:**

N/A

**Review Assessment: Checking Correctness Of Experiments:**

N/A

**Review Assessment: Thoroughness In Paper Reading:**

N/A

---

> ### Author Response · Authors · 2019-11-13
> **To reviewer’s comments**
>
> Thank you for the comments!
>
> To review’s feedback:
>
> - We pay attention to the term “skill discovery” and made it more clear about the connection between prior works and the current work in the revised version. Our method can also be combined with DIAYN to learn the skill-conditioned policy as mentioned in the paper.
>
> - We added both a theoretical connection and new experimental results to compare MISC and the empowerment method in the revised version. In the navigation tasks, we show that our method outperforms the empowerment method.
>
> - An intuition for why I(s_c, s_i) could be superior to I(a, s_i) is that in robotic tasks, the mutual information between the robotic sates, s_c, and the object states, s_i, could be easier to be estimated than the mutual information between the action, a, and the object states, s_i, as shown in Figure 4 in the paper. Therefore, the agent receives a higher MI reward more easily and learns to control s_i more efficiently.
> The context states can be seen as the summary information of the agent’s action and the transition model of the environment, which could be more relevant in terms of estimating the object states in comparison to the agent’s actions.
>
> - VIME and PER are used as described in their original papers.
>
> - We have added an appendix to provide more information about experiment details.
>
> - We also newly evaluated our method on gazebo-based robotic simulations, including the cases when there is no object, a single object of interest, and multiple objects of interests.
> A video showing new experimental results is available at https://youtu.be/l5KaYJWWu70?t=104
> In this experiment, we also compare MISC with two additional baselines, including ICM and empowerment (with state of interest), see Figure 4 in the paper.
>
> - We now mention that the states of interests vs context are given in the revised paper. However, when they are not given. They can also be automatically learned/selected by iterating over all possible combinations.  Afterwards, an optimal combination can be chosen by the user via testing in the task at hand.

---

### Official Review · AnonReviewer1 · 2019-10-24
**Official Blind Review #1**

**Rating:** 3

**Review:**

This paper proposes a self-supervised reinforcement learning approach, Mutual Information-based State-Control (MISC), which maximizes the mutual information between the context states (i.e. robot states) and the states of interest (i.e. states of an object to manipulate). For this, they first split the entire state into two mutually exclusive sets of the context states and the states of interest. Then, the neural discriminator is trained to estimate the (lower-bound of) mutual information between the two states. The (mutual-information) intrinsic reward is computed by the trained neural discriminator, which is used for policy pre-training. Experimental results show that MISC helps to improve the performance of DDPG/SAC and the learned discriminator can be transferred to different environments.


Detailed comments and questions:

- In the paper, the states are represented by only object positions (x, y, z). Is this sufficient? (e.g. velocity is unnecessary?)

- For MISC, the additional assumption is required: the agent should know that which parts of the states are its own controllable state and object's state respectively. Is this additional assumption realistic enough and has it been adopted in other previous works? Is there any way to discriminate robot states and object states automatically?

- Can MISC deal with the problems where the number of objects of interest is more than two? In this case, how can we define mutual information?

- In Eq. (4), T(x_1:N, y_1:N) is assumed to be decomposable into the sum of T(x_t, y_t) / N. Can this make the lower bound (Eq. (3)) arbitrarily loose since the class of functions becomes very limited?

- Detailed experimental setups are missing. e.g. network architecture, hyper-parameters (e.g. I_tran^max), and how they were searched.

- Similarly to the problem of sparse reward, if the robot and the object are far apart and it is difficult to reach the object with random exploration, it would also be difficult to train the mutual information discriminator. How was the discriminator trained? How many time steps were used to train MI discriminator?

- It seems that the MI discriminator learns to estimate the 'proximity' between the robot and the object. Compared to using just a very simple dense reward (e.g. negative L2 distance between the robot and the object), what would be the advantage of using MI discriminator? It would be great to show the comparison between using simple dense reward and MI discriminator for each Push, Pick&Place, and Slide task.

- For the MISC+DIAYN, what if we train the agent using MISC and DIAYN at the same time, instead of pre-training MISC first and fine-tuning DIAYN later?

- It is unclear how MISC-p is performed. Please elaborate on how MISC-p works for prioritization.

- Also, for MISC-r experiments, the weights between the intrinsic reward bonus and the extrinsic reward are not specified in the paper.

- It seems that MISC is beneficial when the robot should get closer to the object for the success of the task. Then, how about the opposite situation? What if the task requires that the robot should 'avoid' the object of interest? Does MISC still work? Is it helpful for the improvement of sample efficiency?

- In order to pre-train the discriminator network, additional (s,a,s') experiences are required, thus it seems difficult to say that it is better for exploration than VIME.

- In section 4.3, what happens if we transfer the learned discriminator to Pick&Place from Push that has a gripper fixed to be closed, rather than the opposite direction (i.e. from Pick&Place to Push)? Does the MISC-t still well work? Can the learned MI discriminator be transferred to different tasks even when the state space is different?



**Experience Assessment:**

I have read many papers in this area.

**Review Assessment: Checking Correctness Of Derivations And Theory:**

N/A

**Review Assessment: Checking Correctness Of Experiments:**

I carefully checked the experiments.

**Review Assessment: Thoroughness In Paper Reading:**

N/A

---

> ### Author Response · Authors · 2019-11-13
> **To reviewer’s comments**
>
> Thank you for the comments!
>
> To review’s questions:
>
> - As the experimental results shows, with position information alone, the agent is able to learn to push or pick up the object, therefore we consider position information alone (without velocity information) is sufficient in our case.
>
> - For MISC, the method needs to know what are the states of interests and what are the context state. While, the states of interest can be any states that users are interested in, such as a part of the robot states or the object states. The context states are some other states, which are different from the states of interest. In robotic tasks, the states of the robot and the object states are normally available [Andrychowicz et al 2018, Plappert et al 2018].
>
> To automatically detect the state of interests and the context states, we can train the agent with random state splits and then chose the combination, which is suitable for the tasks at hand.
>
> References:
> [1] Marcin Andrychowicz, Filip Wolski, Alex Ray, Jonas Schneider, Rachel Fong, Peter Welinder, Bob McGrew, Josh Tobin, OpenAI Pieter Abbeel, and Wojciech Zaremba. Hindsight experience replay. In Advances in Neural Information Processing Systems, pp. 5048–5058, 2017.
> [2] Matthias Plappert, Marcin Andrychowicz, Alex Ray, Bob McGrew, Bowen Baker, Glenn Pow- ell, Jonas Schneider, Josh Tobin, Maciek Chociej, Peter Welinder, et al. Multi-goal reinforce- ment learning: Challenging robotics environments and request for research. arXiv preprint arXiv:1802.09464, 2018.
>
> - Yes, MISC can deal with the case, when there are multiple objects of interest. We added new experiments showing the agent can learn to manipulate two balls. We define the mutual information intrinsic reward as I(S^i_{1}, S^c)+I(S^i_{2}, S^c). The experimental results are shown in the new video at https://youtu.be/l5KaYJWWu70?t=148, where we show that a robot car can learn to manipulate two balls in the same episode.
>
> - Equation (4) is not the mutual information between two trajectories of states. It is an estimation of mutual information between two sets of states. And the states are sampled from the same trajectory. Therefore, we do not need to decompose Equation (4) to evaluate Equation (3).
>
> - We add the experimental details in the Appendix.
>
> - The discriminator is trained along with the policy. For example, in the case that we update the agent 200 times in each epoch, then we also update the MISC 200 times per epoch. For more detailed information, please refer to our code at https://github.com/misc-project/misc
>
> - Compared to the dense reward, with the negative L2 distance between the robot and the object, the robot can only learn to reach the object but will not learn to push or pick up the object because when the robot reaches the object, the negative L2 distance is already zero. However, MISC has the advantage that it not only enables the agent to learn to reach but also learn to push and pick & place.
>
> - If we train the MISC and DIAYN at the same time, the DIAYN reward might be dominant. Subsequently, The agent might not learn to control the states of interests with MISC.
>
> - MISC-p works similarly to PER. The main difference is that MISC-p uses the estimated mutual information quantity as a priority, while PER uses the TD-error as a priority for replay. For more detail on PER, please refer to the original PER paper [Schaul et al 2016].
>
> Reference:
> Tom Schaul, John Quan, Ioannis Antonoglou, and David Silver. Prioritized experience replay. In International Conference on Learning Representations, 2016.
>
> - We first scale the intrinsic and the extrinsic reward between 0 and 1 and then use equal weights for these two rewards.
>
> - For the opposite situation, we can use negative mutual information rewards to encourage the agent to learn to “avoid” some objects.
>
> - The discriminator uses the same amount of (s,a,s') experience as VIME consumes because the discriminator is fixed after pre-training. VIME can only be trained along with the policy. VIME cannot be pre-trained, otherwise, it won’t detect novel states.
>
> - Transfer the learned discriminator from Push to the Pick&Place should still help the agent to learn the pick & place task because the transferred discriminator will help the agent to learn to reach the object at least. As long as the state inputs for the discriminator are the same, then MI discriminator can be transferred among different tasks.

---

### Official Review · AnonReviewer2 · 2019-10-25
**Official Blind Review #2**

**Rating:** 3

**Review:**

The paper paper proposes a mutual information maximization objective for discovering unsupervised robotic manipulation skills. The paper assumes that the state space can be divided into two parts - the state of the robot (“context states”) which is controllable via actions and the state of an object (“states of interest”) which must be manipulated by the robot. Given these two categories of states, the proposed algorithm maximizes a lower bound on the mutual information between the two categories of states such that a policy is learnt that is able to manipulate the object with the robot meaningfully.

I vote for weak reject as (1) the paper makes a strong assumption about the availability of both the robot and object states which is not realistic in typical robotic manipulation applications and (2) the objective in the paper will not work if there is no notion of an “object” or object-state e.g.: this algorithm will not learn skills for a robot trying to control itself; hence, it is not truly a general purpose skill discovery algorithm but rather a skill discovery algorithm specifically meant for robot-object manipulation tasks.

My main concern with the paper is its limited applicability to robotic manipulation tasks with a clear divide between states of interest vs others. The paper does not talk about settings where states of interest are not known, so all of the experiments are based on this strong assumption. It doesn’t seem like a surprising discovery that maximizing the mutual information between the robot state and object state will lead to skills that actually make the robot move the object.

Given that object manipulation is the specific application of interest, the comparison with DIAYN and the combined objective with DIAYN is interesting but little motivation or discussion has been provided in the paper. Can the authors elaborate on why this choice should intuitively be better than the proposed method alone?

The paper does not talk about how these skills can be used as primitive actions by a higher level controller (in a hierarchical RL setup), which would help in demonstrating the usefulness of these skills - e.g.: are these skills sequentially composable such that they can solve a complex task?


**Experience Assessment:**

I have read many papers in this area.

**Review Assessment: Checking Correctness Of Derivations And Theory:**

I assessed the sensibility of the derivations and theory.

**Review Assessment: Checking Correctness Of Experiments:**

I assessed the sensibility of the experiments.

**Review Assessment: Thoroughness In Paper Reading:**

I read the paper at least twice and used my best judgement in assessing the paper.

---

> ### Author Response · Authors · 2019-11-13
> **To reviewer’s comments**
>
> Thank you for the comments!
>
> To reviewer’s concerns:
>
> - First of all, the state of interest does not have to be the object state. It can be the state of the robot, for example, the state of actuators. Maximizing the mutual information between two sets of actuator states can help the agent to learn to control itself. We did a new experiment in navigation environments, where train the agent to maximize the mutual information between its left wheel states and its right wheel states. The agent learns to run in a straight line instead of in random directions. The video showing experiment results is available at https://youtu.be/l5KaYJWWu70?t=134
>
> - Although we evaluated our method in robotic manipulation tasks, it does not mean it won’t work for other tasks. We added additional experiments in a new navigation task, see the video at https://youtu.be/l5KaYJWWu70?t=104
> We consider our algorithm as a general-purpose skill learning algorithm in the sense that it guides the agent to learn any skills to control the states of interests. The states of interest could be any states, such as the robot states, the object states, or the states of the environment.
>
> - The state of interest is specified by the user with little domain knowledge. However, when there is no clear divide from the user, the agent can learn from different combinations of the states of interest and the context states. In the end, the user can choose skills from the learned skill sets that are useful for the task at hand.
>
> - The combination of our method and DIAYN enables DIAYN to learn manipulation skills efficiently, while DIAYN alone did not learn. Furthermore, compared to MISC, the combined method enjoys the benefits brought by DIAYN, such as learning combinable motion primitive with skill-conditioned policy for hierarchical reinforcement learning [1].
>
> Reference:
> [1] Benjamin Eysenbach, Abhishek Gupta, Julian Ibarz, and Sergey Levine. Diversity is all you need: Learning skills without a reward function. In International Conference on Learning Representations, 2019. URL https://openreview.net/forum?id=SJx63jRqFm.

---

### Author Response · Authors · 2019-11-13
**Revision**


- We did new experiments in a navigation task, see Figure 1, with three different settings including a single object, no object, and multiple objects.
- A new video is uploaded to show the new experimental results at https://youtu.be/l5KaYJWWu70?t=104
- We also added a theoretical proof of the connection between our method MISC and the empowerment method in the Appendix.
- We added new experiments in the navigation task to compare MISC, empowerment and ICM methods, see Figure 4 in the paper.
- Experimental details are now added in the Appendix.
- We updated the paper about the new experiments mentioned above.
- We updated the paper according to review comments.

---

### Decision · Program_Chairs · 2019-12-19

**Decision:**

Reject

**Comment:**

The paper considers a setting where the state of a (robotics) environment can be divided roughly into "context states" (such as variables under the robot's direct control) and "states of interest" (such as the state variables of an object to be manipulated), and learn skills by maximizing a lower bound on the mutual information between these two components of the state. Experimental results compare to DDPG/SAC, and show that the learned discriminator is somewhat transferable between environments.

Reviewers found the assumptions necessary on the degree of domain knowledge to be quite strong and domain-specific, and that even after revision, the authors were understating the degree to which this was necessary. The paper did improve based on reviewer feedback, and while R3 was more convinced by the follow-up experiments (though remarked that requiring environment variations to obtain new skills was a "significant step backward from things like [Diversity is All You Need]"), the other reviewers remained unconvinced regarding domain knowledge and in particular how it interacts with the scalability of the proposed method to complex environments/robots.

Given the reviewers' concerns regarding applicability and scalability, I recommend rejection in its present form. A future revision may be able to more convincingly demonstrate that limitations based on domain knowledge are less significant than they appear.